# Graph Classification with 2D Convolutional Neural Networks

## Abstract

Graph classification is currently dominated by graph kernels, which, while powerful, suffer some significant limitations. Convolutional Neural Networks (CNNs) offer a very appealing alternative. However, processing graphs with CNNs is not trivial. To address this challenge, many sophisticated extensions of CNNs have recently been proposed. In this paper, we reverse the problem: rather than proposing yet another graph CNN model, we introduce a novel way to represent graphs as multi-channel image-like structures that allows them to be handled by vanilla 2D CNNs. Despite its simplicity, our method proves very competitive to state-of-the-art graph kernels and graph CNNs, and outperforms them by a wide margin on some datasets. It is also preferable to graph kernels in terms of time complexity. Code and data are publicly available[1].

## 1 Graph classification

Graphs, or networks, are rich, flexible, and universal structures that can accurately represent the interaction among the components of many natural and human-made complex systems (Dorogovtsev & Mendes, 2013). For instance, graphs have been used to describe and analyze the interplay among proteins within cells and the internal structure of proteins themselves (Borgwardt et al., 2007; 2005), the organization of the brain (Bullmore & Sporns, 2009), the World Wide Web (Page et al., 1999), textual documents (Mihalcea & Tarau, 2004), and information propagation through a population (Kitsak et al., 2010). Consequently, graph mining has attracted significant attention in machine learning and artificial intelligence, and is still today a very active area of investigation. A central graph mining task is that of *graph classification* (not to be mistaken with *node* classification). Its applications range from determining whether a protein is an enzyme or not in bioinformatics to categorizing documents in NLP and analyzing a social network. Graph classification is the task of interest in this study.

## 2 Limitations of graph kernels

The state-of-the-art in graph classification is currently dominated by a family of methods referred to as *graph kernels*. Graph kernels compute the similarity between two graphs as the sum of the pairwise similarities between some of their substructures, and then pass the similarity matrix computed on the entire dataset to a kernel-based supervised algorithm such as the Support Vector Machine (Cortes & Vapnik, 1995) to learn soft classification rules. Graph kernels mainly vary based on the substructures they use, which include random walks (Gärtner et al., 2003), shortest paths (Borgwardt & Kriegel, 2005), and subgraphs (Shervashidze et al., 2009), to cite only a few. While graph kernels have been very successful, they suffer significant limitations:

1. **High time complexity**. This problem is threefold: first, populating the kernel matrix requires computing the similarity between every two graphs in the training set (say of size $N$), which amounts to $N(N-1)/2$ operations. The cost of training therefore increases much more rapidly than the size of the dataset. Second, computing the similarity between a pair of graphs (i.e., performing a single operation) is itself polynomial in the number of nodes. For instance, the time complexity of the

---

[1]link will be provided upon acceptance

shortest path graph kernel is $\mathcal{O}(|V_1|^2|V_2|^2)$ for two graphs $(V_1, V_2)$, where $|V_i|$ is the number of nodes in graph $V_i$. Processing large graphs can thus become prohibitive, which is a serious limitation as big networks abound in practice. Finally, finding the support vectors is $\mathcal{O}(N^2)$ when the $C$ parameter of the SVM is small and $\mathcal{O}(N^3)$ when it gets large (Bottou & Lin, 2007), which can again pose a problem on big datasets.

2. **Disjoint feature and rule learning**. With graph kernels, the computation of the similarity matrix and the learning of the classification rules are two independent steps. In other words, the features are fixed and not optimized for the task.

3. **Graph comparison is based on small independent substructures**. As a result, graph kernels focus on local properties of graphs, ignoring their global structure (Nikolentzos et al., 2017). They also underestimate the similarity between graphs and suffer unnecessarily high complexity (due to the explosion of the feature space), as substructures are considered to be orthogonal dimensions (Yanardag & Vishwanathan, 2015).

## 3 PROPOSED METHOD

### 3.1 OVERVIEW

We propose a very simple approach to turn a graph into a multi-channel image-like structure suitable to be processed by a traditional 2D CNN. It can be broken down into 3 steps, summarized in Figure 1. The first step involves embedding the nodes of the graph. The embedding space is compressed with PCA at step 2. We then repeatedly extract 2D slices from the compressed space and compute a 2D histogram for each slice. The "image" representation of the graph is finally given by the stack of its 2D histograms (each histogram making for a channel). Note that the dimensionality of the final representation of a graph does not depend on its number of nodes or edges. Big and small graphs are represented by images of the same size.

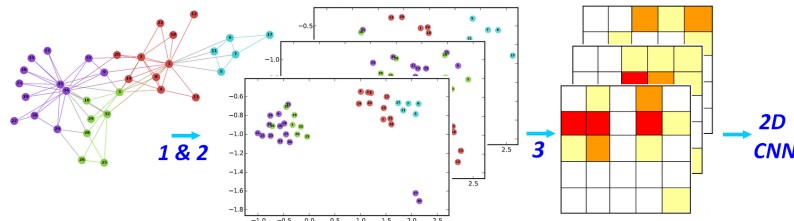

Figure 1: Our 3-step approach represents graphs as "images" suitable to be passed to vanilla 2D CNNs. Steps 1 & 2: graph node embeddings and compression with PCA Step 3: computation and stacking of the 2D histograms.

Our method addresses the limitations of graph kernels in the following ways:

1. **High time complexity**. By converting all graphs in a given dataset to representations of the same dimensionality, and by using a classical 2D CNN architecture for processing those graph representations, our method offers constant time complexity at the instance level, and linear time complexity at the dataset level. Moreover, state-of-the-art node embeddings can be obtained for a given graph in linear time (w.r.t. the size of the graph), for instance with `node2vec` (Grover & Leskovec, 2016).

2. **Disjoint feature and rule learning**. Thanks to the 2D CNN classifier, features are learned directly from the raw data during training to optimize performance on the downstream task.

3. **Graph comparison is based on small independent substructures**. Our approach capitalizes on state-of-the-art graph node embedding techniques that capture both local and global properties of graphs. In addition, we remove the need for handcrafted features.

### 3.2 DETAILS

#### 3.2.1 THE NEED FOR SPATIAL DEPENDENCE

Convolutional Neural Networks (CNNs) are feedforward neural networks specifically designed to work on regular grids. A regular grid is the $d$-dimensional Euclidean space discretized by parallelo-

topes (rectangles for $d = 2$, cuboids for $d = 3$, etc.). In CNNs, each neuron in a given layer receives input from a neighborhood of the neurons in the previous layer (LeCun et al., 1998). Those neighborhoods, or *local receptive fields*, allow CNNs to compose higher-level features from lower-level features, and thus to capture patterns of increasing complexity in a hierarchical way. Regular grids satisfy the *spatial dependence*[2] property, which is the fundamental premise on which local receptive fields and hierarchical composition of features in CNNs hold.

### 3.2.2 HOW TO REPRESENT GRAPHS AS STRUCTURES THAT VERIFY THE SPATIAL DEPENDENCE PROPERTY?

Traditionally, a graph $G(V, E)$ is encoded as its adjacency matrix $A$ or Laplacian matrix $L$. $A$ is a square matrix of dimensionality $|V| \times |V|$, symmetric in the case of undirected graphs, whose $(i, j)^{th}$ entry $A_{i,j}$ is equal to the weight of the edge $e_{i,j}$ between nodes $v_i$ and $v_j$, if such an edge exists, or to 0 otherwise. On the other hand, the Laplacian matrix $L$ is equal to $D - A$, where $D$ is the diagonal degree matrix. One could initially consider passing one of those structures as input to a 2D CNN. However, unlike in images, where close pixels are more strongly correlated than distant pixels, adjacency and Laplacian matrices are not associated with spatial dimensions and the notion of Euclidean distance, and thus do not satisfy the spatial dependence property. As will be detailed next, we capitalize on *graph node embeddings* to address this issue.

**Step 1: Graph node embeddings**. There is local correlation in the node embedding space. In that space, the Euclidean distance between two points is meaningful: it is inversely proportional to the similarity of the two nodes they represent. For instance, two neighboring points in the embedding space might be associated with two nodes very distant in the graph, but playing the same structural role (e.g., of flow control), belonging to the same community, or sharing some other common property.

**Step 2: Alignment and compression with PCA**. As state-of-the-art node embedding techniques (such as `node2vec`) are neural, they are stochastic. Dimensions are thus recycled from run to run, meaning that a given dimension will not be associated with the same latent concepts across graphs, or across several runs on the same graph. Therefore, to ensure that the embeddings of all the graphs in the collection are comparable, we apply PCA and retain the first $d \ll D$ principal components (where $D$ is the dimensionality of the original node embedding space). PCA also serves an information maximization (compression) purpose. Compression is desirable as it greatly reduces the shape of the tensors fed to the CNN (for reasons that will become clear in what follows), and thus complexity, at the expense of a negligible loss in information.

**Step 3: Computing and stacking 2D histograms**. We finally repeatedly extract 2D slices from the $d$-dimensional PCA node embedding space, and turn those planes into regular grids by discretizing them into a finite, fixed number of equally-sized bins, where the value associated with each bin is the count of the number of nodes falling into that bin. In other words, we represent a graph as a stack of $d/2$ 2D histograms of its (compressed) node embeddings[3]. As illustrated in Figure 2, the first histogram is computed from the coordinates of the nodes in the plane made of the first two principal directions, the second histogram from directions 3 and 4, and so forth. Note that using adjacent and following PCA dimensions is an arbitrary choice. It ensures at least that channels are sorted according to the amount of information they contain.

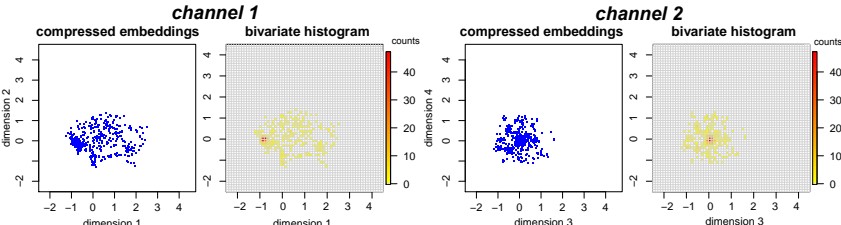

Figure 2: Node embeddings and image representation of graph ID #10001 (577 nodes, 1320 edges) from the REDDIT-12K dataset.

---

[2] the concept of spatial dependence is well summarized by: "everything is related to everything else, but near things are more related than distant things" (Tobler, 1970). For instance in images, close pixels are more related than distant pixels.

[3] our representation is unrelated to the widespread *color histogram* encoding of images.

Using computer vision vocabulary, bins can be viewed as *pixels*, and the 2D slices of the embedding space as *channels*. However, in our case, instead of having 3 channels (R,G,B) like with color images, we have $d/2$ of them. That is, each pixel (each bin) is associated with a vector of size $d/2$, whose entries are the counts of the nodes falling into that bin in the corresponding 2D slice of the embedding space. Finally, the *resolution* of the image is determined by the number of bins of the histograms, which is constant for a given dataset across all dimensions and channels.

## 4 EXPERIMENTAL SETUP

### 4.1 2D CNN ARCHITECTURE

We implemented a variant of LeNet-5 (LeCun et al., 1998) with which we reached 99.45% accuracy on the MNIST handwritten digit classification dataset. As illustrated in Figure 3 for an input of shape (5,28,28), this simple architecture deploys four convolutional-pooling layers (each repeated twice) in parallel, with respective region sizes of 3, 4, 5 and 6, followed by two fully-connected layers. Dropout (Srivastava et al., 2014) is employed for regularization at every hidden layer. The activations are `ReLU` functions (in that, our model differs from LeNet-5), except for the ultimate layer, which uses a `softmax` to output a probability distribution over classes. For the convolution-pooling block, we employ 64 filters at the first level, and as the signal is halved through the (2,2) max pooling layer, the number of filters in the subsequent convolutional layer is increased to 96 to compensate for the loss in resolution.

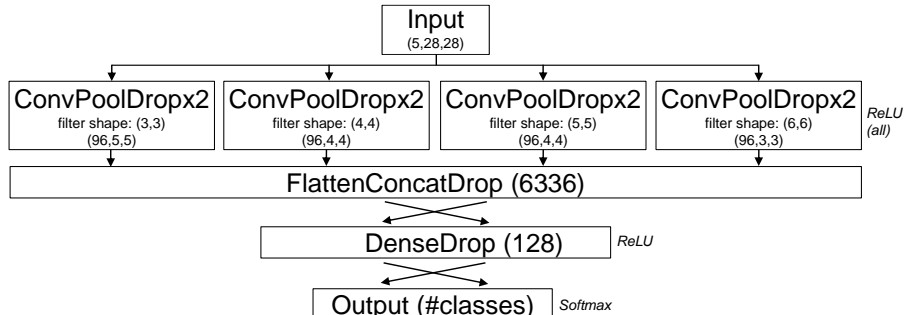

Figure 3: 2D CNN architecture used in our experiments. The number within parentheses refer to the *output* dimensions of the tensors.

### 4.2 NEURAL EMBEDDINGS

We used `node2vec` (Grover & Leskovec, 2016), which applies the very fast Skip-Gram language model (Mikolov et al., 2013) to truncated biased random walks performed on the graph. `node2vec` scales linearly with the number of nodes in the network. We leveraged the `igraph` module (Csardi & Nepusz, 2006) and the publicly available high performance C++ implementation[4] of `node2vec`.

### 4.3 DATASETS

We conducted experiments on the publicly available[5] datasets from (Yanardag & Vishwanathan, 2015) which we briefly describe in what follows, and in Table 1. In all datasets, graphs are unweighted, undirected, with unlabeled nodes, and the task is to predict the class they belong to. Classes are mutually exclusive. REDDIT-B, REDDIT-5K, and IMDB-B are perfectly balanced, whereas REDDIT-12K and COLLAB feature a maximum class imbalance ratio of 1:5 and 1:3.4, respectively. Note that graphs with less than 10 nodes were removed because having 5 channels requires at least a 10-dimensional embedding space, which is impossible to obtain with less than 10 nodes. However, this represented only a few graphs per dataset, for some datasets.

---

[4] `https://github.com/snap-stanford/snap/tree/master/examples/node2vec`
[5] `http://www.mit.edu/~pinary/kdd/datasets.tar.gz`

| Dataset | IMDB-B | COLLAB | REDDIT-B | REDDIT-5K | REDDIT-12K |
|---|---|---|---|---|---|
| Max # vertices | 136 | 492 | 3782 | 3648 | 3782 |
| Min # vertices | 12 | 32 | 6 | 22 | 2 |
| Average # vertices | 19.77 | 74.49 | 429.61 | 508.50 | 391.40 |
| Max # edges | 1249 | 40120 | 4071 | 4783 | 5171 |
| Min # edges | 26 | 60 | 4 | 21 | 1 |
| Average # edges | 96.53 | 2457.78 | 497.75 | 594.87 | 456.89 |
| # graphs | 1000 | 5000 | 2000 | 4999 | 11929 |
| # classes | 2 | 3 | 2 | 5 | 11 |
| Max class imbalance | 1:1 | 1:3.4 | 1:1 | 1:1 | 1:5 |

Table 1: Statistics of the social network datasets used in our experiments.

In all REDDIT datasets, a graph corresponds to a thread where nodes represent users, and there is an edge between two nodes if one of the two users responded to a comment from the other user. More precisely, graphs in **REDDIT-B** are labeled according to whether they were constructed from Q&A communities or discussion communities, and **REDDIT-5K** and **REDDIT-12K** respectively feature graphs from 5 and 11 forums dedicated to specific topics (those forums are known as "subreddits"). In **COLLAB**, graphs are hop-1 neighborhoods of researchers from a scientific collaboration network (two researchers are linked if they co-authored a paper), and are labeled according to the subfield of Physics the corresponding researcher belongs to. Finally, the **IMDB-B** dataset features hop-1 neighborhoods of actors and actresses selected from two movie collaboration networks corresponding to specific genres (action and romance), in which two actors are linked if they starred in the same movie. Graphs are labeled according to the genre they were sampled from. We refer the reader to the original paper for more information about the datasets.

## 4.4 BASELINES

We compared our model to two state-of-the-art graph kernels, the graphlet kernel (Shervashidze et al., 2009) and the Weisfeiler-Lehman (WL) kernel (Shervashidze et al., 2011).
The **graphlet** kernel computes the similarity between two graphs as the cosine of their count vectors. These vectors encode how many subgraphs of size up to a certain threshold can be found in each graph (each entry is an occurrence count). We sampled 2000 graphlets of size up to 6 from each graph.
The **WL** kernel is actually a framework that operates on top of any graph kernel accepting node labels and boosts its performance by using the relabeling procedure of the WL test of isomorphism. More precisely, following the computation of the kernel value between the two graphs, vertex labels are updated based on the labels of their neighbors. This two-step process repeats for a certain number of iterations. The final kernel value is the sum of the values at each iteration. Since our graphs have unlabeled nodes, we set the degrees of the nodes as their labels. Furthermore, we used the WL framework with the subtree graph kernel (Gärtner et al., 2003), as it is very efficient with this kernel (Shervashidze et al., 2011).
For both baselines, we used a C-SVM classifier[6] (Pedregosa et al., 2011). The C parameter of the SVM and the number of iterations in WL were jointly optimized on a 90-10 % partition of the training set of each fold by searching the grid $\left\{(10^{-4}, 10^4, \text{len} = 10); (2, 7, \text{step} = 1)\right\}$.

## 4.5 CONFIGURATION

All experiments involved 10-fold cross validation where each fold was repeated 3 times. We used Xavier initialization (Glorot & Bengio, 2010), a batch size of 32, and for regularization, a dropout rate of 0.3 and early stopping with a patience of 5 epochs. The categorical cross-entropy loss was optimized with Adam (Kingma & Ba, 2014) (default settings). We implemented our model in `Keras` (Chollet et al., 2015) version 1.2.2[7] with `tensorflow` (Abadi et al., 2016) backend. The hardware used consisted in an NVidia Titan Xp GPU with an 8-thread Intel Xeon 2.40 GHz CPU and 16 GB of RAM, under Ubuntu 16.04.2 LTS 64-bit operating system and Python 2.7. The graph kernel

---

[6]http://scikit-learn.org/stable/modules/generated/sklearn.svm.SVC.html
[7]https://faroit.github.io/keras-docs/1.2.2/

baselines were run on an 8-thread Intel i7 3.4 GHz CPU, with 16 GB of RAM, under Ubuntu 16.06 LTS 64-bit operating system and Python 2.7.

### 4.6 RESOLUTION, CHANNELS, AND NODE2VEC PARAMETERS

In our initial experiments involving spectral embeddings, the coordinates of any node in any dimension belonged to the $[-1, 1]$ range, due to the eigenvectors being unit-normed. Furthermore, inspired by the MNIST images which are $28 \times 28$ in size, and on which we initially tested our 2D CNN architecture, we decided to learn 2D histograms featuring 28 bins in each direction. This gave us a resolution of $28/(1-(-1))$, that is, 14 pixels per unit (or simply 14:1). As it was giving good results, we stuck to similar values in our final experiments making use of neural embeddings.

With the $p$ and $q$ parameters of `node2vec` held constant and equal to 1, we conducted a search on the coarse grid $\{(14,9);(2,5)\}$ to get more insights about the impact of resolution and number of channels (respectively). On a given dataset, image size is calculated as the range $|max(\text{coordinates}) - min(\text{coordinates})|$ multiplied by the resolution, where "coordinates" are the node loadings flattened across all dimensions of the embedding space. For instance, on COLLAB with a resolution of 9:1, image size is equal to $37 \times 37$, since $|2.78 - (-1.33)| \times 9 \approx 37$. Optimal values for each dataset are summarized in Table 2.

With the best resolution and number of channels, we then tuned the return and in-out parameters $p$ and $q$ of `node2vec`. Those parameters respectively bias the random walks towards exploring larger areas of the graph or staying in local neighborhoods, allowing the embeddings to encode a similarity that interpolates between structural equivalence (two nodes acting as, e.g., flow controllers, are close to each other) and homophily (two nodes belonging to the same community are close to each other). We tried 5 combinations of values for $(p, q)$: $\{(1, 1); (0.25, 4); (4, 0.25); (0.5, 2); (2, 0.5)\}$. Note that $p = q = 1$ is equivalent to `DeepWalk` (Perozzi et al., 2014).

| | REDDIT-B | REDDIT-5K | REDDIT-12K | COLLAB | IMDB-B |
|---|---|---|---|---|---|
| Res. | 9:1 | 9:1 | 9:1 | 9:1 | 14:1 |
| #Chann. | 5 | 2 | 5 | 5 | 5 |
| p,q | 2,0.5 | 4,0.25 | 1,1 | 0.25,4 | 1,1 |

Table 2: Best resolution, number of channels, and $(p, q)$ for each dataset.

## 5 RESULTS

The classification accuracy of our approach and the baselines we implemented are reported in Table 3. Even though we did not re-implement those models, we also display for comparison purposes the performance reported in Yanardag & Vishwanathan (2015) (Deep Graph Kernels) and Niepert et al. (2016) (Graph CNN, PSCN $k = 10$), since the experimental setting is the same.

Our approach shows significantly better than all baselines on the REDDIT-12K and REDDIT-B datasets, with large improvements of 6.81 and 2.82 in accuracy over the best performing competitor, respectively. We also reach best performance on the REDDIT-5K dataset, with an improvement in accuracy of 1.34 over the best performing baseline. However, the difference is not statistically significant. Finally, on the IMDB-B dataset, we get third place, very close ($\leq 1.2$) to the top performers, and again without statistically significant differences. Actually, the only dataset on which a baseline proved significantly better than our approach is COLLAB (WL graph kernel).

### 5.1 RUNTIMES

Even if not directly comparable, we report in Table 4 kernel matrix computation time for the two graph kernel baselines, along with the time required by our 2D CNN model to perform one pass over the entire training set, i.e., the time per epoch. With respects to time complexity, our method is superior to graph kernels on several counts: first, unlike graph kernels, the time required by the 2D CNN to process one training example is constant (all images for a given dataset have the same size), while computing the kernel value for a pair of graphs depends on their size (polynomial in the number of nodes). It is true that a prerequisite for our approach is an embedding for all the graphs in the dataset, but `node2vec` scales linearly with the number of nodes in the graph. Therefore, on

big graphs, our method is still usable, while graph kernels may not be. Also, `node2vec` is easily parallelizable over the collection, so one can take advantage of multi-core CPUs to considerably speed up the process. Second, with a 2D CNN, the time necessary to go through the entire training set only increases linearly with the size of the set, while populating the kernel matrix is quadratic, and finding the support vectors is then again at least quadratic. This means that on large datasets, our approach is also preferable to graph kernels. Examples of 2D CNN architectures much more complex than ours applied to millions of images in reasonable time abound in the recent computer vision literature. Processing such big datasets with graph kernels would simply be intractable. In addition, not only do neural models allow processing big datasets, but their performance also tends to significantly improve in the presence of large quantities of training data.

Table 3: 10-fold CV average test set classification accuracy of our proposed method compared to state-of-the-art graph kernels and graph CNN. $\pm$ is standard deviation. Best performance per column in **bold**. $^\star$indicates stat. sign. at the $p < 0.05$ level (our 2D CNN vs. WL) using the Mann-Whitney U test (`https://docs.scipy.org/doc/scipy-0.19.0/reference/generated/scipy.stats.mannwhitneyu.html`).

| Dataset
Method | REDDIT-B
(size=2,000;nclasses=2) | REDDIT-5K
(4,999;5) | REDDIT-12K
(11,929;11) | COLLAB
(5,000;3) | IMDB-B
(1,000;2) |
|---|---|---|---|---|---|
| Graphlet Shervashidze2009 | 77.26 ($\pm$ 2.34) | 39.75 ($\pm$ 1.36) | 25.98 ($\pm$ 1.29) | 73.42 ($\pm$ 2.43) | 65.40 ($\pm$ 5.95) |
| WL Shervashidze2011 | 78.52 ($\pm$ 2.01) | 50.77 ($\pm$ 2.02) | 34.57 ($\pm$ 1.32) | **77.82**$^\star$ ($\pm$ 1.45) | **71.60** ($\pm$ 5.16) |
| Deep GK Yanardag2015 | 78.04 ($\pm$ 0.39) | 41.27 ($\pm$ 0.18) | 32.22 ($\pm$ 0.10) | 73.09 ($\pm$ 0.25 ) | 66.96 ($\pm$ 0.56 ) |
| PSCN $k = 10$ Niepert2016 | 86.30 ($\pm$ 1.58) | 49.10 ($\pm$ 0.70) | 41.32 ($\pm$ 0.42) | 72.60 ($\pm$ 2.15) | 71.00 ($\pm$ 2.29) |
| 2D CNN (our method) | **89.12**$^\star$ ($\pm$ 1.70) | **52.11** ($\pm$ 2.24) | **48.13**$^\star$ ($\pm$ 1.47) | 70.28 ($\pm$ 1.21) | 70.40 ($\pm$ 3.85) |

| | REDDIT-B | REDDIT-5K | REDDIT-12K | COLLAB | IMDB-B |
|---|---|---|---|---|---|
| Size, average (# nodes, # edges) | 2000, (430,498) | 4999, (509,595) | 11929, (391,457) | 5000, (74,2458) | 1000, (20,97) |
| Input shapes (for our approach) | (5,62,62) | (2,65,65) | (5,73,73) | (5,36,36) | (5,37,37) |
| Graphlet Shervashidze2009 | 551 | 5,046 | 12,208 | 3,238 | 275 |
| WL Shervashidze2011 | 645 | 5,087 | 20,392 | 1,579 | 23 |
| 2D CNN (our approach) | 6 | 16 | 52 | 5 | 1 |

Table 4: Runtimes in seconds, rounded to the nearest integer. For the graph kernel baselines, time necessary to populate the Kernel matrix (8-thread 3.4GHz CPU). For our model, time per epoch (Titan Xp GPU).

## 6 RELATED WORK

Motivated by the outstanding performance recently reached by Convolutional Neural Networks (CNNs) in computer vision, e.g. (Vinyals et al., 2015; Krizhevsky et al., 2012), many research efforts have been devoted to generalizing CNNs to graphs. Indeed, CNNs offer a very appealing alternative to kernel-based methods. The parsimony achieved through weight sharing makes them very efficient, their time complexity is constant for each training example and linear with respect to the size of the dataset, and the extra expressiveness they bring might translate to significant accuracy gains.

However, since convolution and pooling are natively defined for regular, low-dimensional grids such as images (2D Euclidean space discretized by rectangles), generalizing CNNs to graphs, which are irregular, non-Euclidean objects, is far from trivial. Possible solutions that can be found in the literature fall into two broad categories: *spatial* and *spectral* techniques (Bruna et al., 2013). Spectral approaches (Defferrard et al., 2016; Kipf & Welling, 2016) invoke the convolution theorem from signal processing theory to perform graph convolutions as pointwise multiplications in the Fourier domain of the graph. The basis used to send the graph to the Fourier domain is given by the SVD decomposition of the Laplacian matrix of the graph, whose eigenvalues can be viewed as "frequencies". By contrast, spatial methods (Niepert et al., 2016; Vialatte et al., 2016) operate directly on the graph structure. For instance, in (Niepert et al., 2016), the algorithm first determines the sequence of nodes for which neighborhood graphs (of equal size) are created. To serve as receptive fields, the neighborhood graphs are then normalized, i.e., mapped to a vector space with a linear order, in which nodes with similar structural roles in the neighborhood graphs are close to each other. Normalization is the central step, and is performed via a labeling procedure. A 1D CNN architecture is finally applied to the receptive fields.

### 6.1 DEPARTURE FROM PREVIOUS WORK

While the aforementioned sophisticated frameworks have made great strides, we showed in this paper that graphs can also be processed by vanilla 2D CNN architectures. This is made possible by the novel graph representation we introduce, which encodes graphs as stacks of 2D histograms of their node embeddings. Compared to the more complex approaches that involve different fundamental architectural and/or operational modifications, the main advantage of our method is its simplicity. Crucially, we show that this simplicity can be obtained *without giving up accuracy*: we indeed outperform graph CNN baselines by a wide margin on some datasets, and are very close elsewhere.

### 6.2 DISCUSSION

Replacing the raw counts by the empirical joint probability density function, either by normalizing the histograms, or with a Kernel Density Estimate, significantly deteriorated performance. This suggests that keeping the absolute values of the counts is important, which makes sense, because some categories might be associated with larger or smaller graphs, on average. Therefore, preventing the model from using size information is likely to decrease accuracy. We also observed that increasing the number of channels to more than 5 does not yield better results (which makes sense, as channels contain less and less information), but that reducing this number improves performance in some cases, probably because it plays a regularization role.

The main contribution of our study is a novel method for *representing* graphs as multi-channel image-like structures from their node embeddings, that allows them to be processed by 2D CNNs. How the embeddings are computed, and which 2D CNN architecture is used, does not matter. We hold this flexibility to be a major strength. First, the *embedding-agnostic* nature of our method means that it can be seamlessly extended to directed, weighted, or labeled graphs with continuous or categorical node/edge attributes, simply by using an embedding algorithm that accepts such graphs, e.g., (Liao et al., 2017). The independence of our approach with respect to the image classification model used is another advantage. Here, we employed a vanilla 2D CNN architecture as it was offering an excellent trade-off between accuracy and simplicity, but more recent models, such as the one of Huang et al. (2016), may yield even better results. Above all, performance should improve as graph node embedding algorithms and CNN architectures for images improve in the future.

Even though results are very good out-of-the-box in most cases, finding an embedding algorithm that works well, or the right combination of parameters for a given dataset, can require some efforts. For instance, on COLLAB, we hypothesize that our results are inferior to that observed on the other datasets because optimizing $p$ and $q$ for COLLAB may require more than a coarse grid search, or because `node2vec` may not be well-suited to very dense graphs such as the ones found in COLLAB.

## 7 CONCLUSION

The main contribution of this paper is to show that CNN architectures designed for images can be used for graph processing in a completely off-the-shelf manner, simply by representing graphs as stacks of two-dimensional histograms of their node embeddings. Despite the simplicity of our approach, results indicate that it is very competitive to state-of-the-art graph kernels and graph CNN models, sometimes outperforming them by a wide margin. Furthermore, these good results were obtained with limited parameter tuning and by using a basic 2D CNN model. From a time complexity perspective, our approach is preferable to graph kernels too, allowing to process larger datasets featuring bigger graphs.

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
