# OpenReview forum: "Graph Classification with 2D Convolutional Neural Networks"
_ICLR.cc/2018/Conference — Reject_

### Official Review · AnonReviewer3 · 2017-11-27
**Minor technical contribution, experiments are not convincing**

**Rating:** 3
**Confidence:** 5

**Review:**

The authors propose to use 2D CNNs for graph classification by transforming graphs to an image-like representation from its node embedding. The approach uses node2vec to obtain a node embedding, which is then compacted using PCA and turned into a stack of discretized histograms. Essentially the authors propose an approach to use a node embedding to achieve graph classification.

In my opinion there are several weak points:

1) The approach to obtain the image-like representation is not well motivated. Other approaches how to  aggregate the set of node embeddings for graph classification are known, see, e.g., "Representation Learning on Graphs: Methods and Applications", William L. Hamilton, Rex Ying, Jure Leskovec, 2017. The authors should compare to such methods as a baseline.

2) The experimental evaluation is not convincing:
- the selection of competing methods is not sufficient. I would like to suggest to add an approach similar to Duvenaud et al., "Convolutional networks on graphs for learning molecular fingerprints", NIPS 2015.
- the accuracy results are taken from other publications and it is not clear that this is an authoritative comparison; the accuracy results published for state-of-the-art graph kernels are superior to those obtained by the proposed method, cf., e.g., Kriege et al., "On Valid Optimal Assignment Kernels and Applications to Graph Classification", NIPS 2016.
- it would be interesting to apply the approach to graphs with discrete and continuous labels.

3) The authors argue that their method is preferable to graph kernels in terms of time complexity. This argument is questionable. Most graph kernels compute explicit feature maps and can therefore be used with efficient linear SVMs (unfortunately most publications use a kernelized SVM). Moreover, the running of computing the node embedding must be emphasized: On page 2 the authors claim a "constant time complexity at the instance level", which is not true when also considering the running time of node2vec. Moreover, I do not think that node2vec is more efficient than, e.g., Weisfeiler-Lehman refinement used by graph kernels.

In summary: Since the technical contribution is limited, the approach needs to be justified by an authoritative experimental comparison. This is not yet achieved with the results presented in the submitted paper. Therefore, it should not be accepted in its current form.

---

### Official Review · AnonReviewer2 · 2017-11-27
**Simple ad-hoc idea not well evaluated**

**Rating:** 4
**Confidence:** 3

**Review:**

The paper introduces a method for learning graph representations (i.e., vector representations for graphs). An existing node embedding method is used to learn vector representations for the nodes. The node embeddings are then projected into a 2-dimensional space by PCA. The 2-dimensional space is binned using an imposed grid structure. The value for a bin is the (normalized) number of nodes falling into the corresponding region.

The idea is simple and easily explained in a few minutes. That is an advantage. Also, the experimental results look quite promising. It seems that the methods outperforms existing methods for learning graph representations.

The problem with the approach is that it is very ad-hoc. There are several (existing) ideas of how to combine node representations into a representation for the entire graph. For instance, averaging the node embeddings is something that has shown promising results in previous work. Since the methods is so ad-hoc (node2vec -> PCA -> discretized density map -> CNN architecure) and since a theoretical understanding of why the approach works is missing, it is especially important to compare your method more thoroughly to simpler methods. Again, pooling operations (average, max, etc.) on the learned node2vec embeddings are examples of simpler alternatives.

The experimental results are also not explained thoroughly enough. For instance, since two runs of node2vec will give you highly varying embeddings (depending on the initialization), you will have to run node2vec several times to reduce the variance of your resulting discretized density maps. How many times did you run node2vec on each graph?

---

### Official Review · AnonReviewer1 · 2017-11-27
**A well written, clear paper presenting a novel representation of graphs   as multi-channel image-like structures from their node embeddings.**

**Rating:** 7
**Confidence:** 3

**Review:**

The paper presents a novel representation of graphs as multi-channel image-like structures. These structures are extrapolated  by
1) mapping the graph nodes into an embedding using an algorithm like node2vec
2) compressing the embedding space using pca
3) and extracting 2D slices from the compressed space and computing 2D histograms per slice.
he resulting multi-channel image-like structures are then feed into vanilla 2D CNN.

The papers is well written and clear, and proposes an interesting idea of representing graphs as multi-channel image-like structures. Furthermore, the authors perform experiments with real graph datasets from the social science domain and a comparison with the SoA method both graph kernels and deep learning architectures. The proposed algorithm in 3 out of 5 datasets, two of theme with statistical significant.

---

### Decision · Program_Chairs · 2018-01-29
**ICLR 2018 Conference Acceptance Decision**

**Decision:**

Reject

**Comment:**

The submission proposes a strategy for creating vector representations of graphs, upon which a CNN can be applied.  Although this is a useful problem to solve, there are multiple works in the existing literature for doing so.  Given that the choice between these is essentially empirical, a through comparison is necessary.  This was pointed out in the reviews, and relevant missing comparisons were given.  The authors did not provide a response to these concerns.